# Non-Uniform Noise Injection for Enhancing DNN Adversarial Robustness and Efficiency

## Abstract

Deep Neural Networks (DNNs) have revolutionized a wide range of industries, from healthcare and finance to automotive, by offering unparalleled capabilities in data analysis and decision-making. Despite their transforming impact, DNNs face two critical challenges: the vulnerability to adversarial attacks and the increasing computational costs associated with more complex and larger models. In this paper, we introduce an effective method designed to simultaneously enhance adversarial robustness and execution efficiency. Unlike prior studies that enhance robustness via uniformly injecting noise, we introduce a non-uniform noise injection algorithm, strategically applied at each DNN layer to disrupt adversarial perturbations introduced in attacks. By employing approximation techniques, our approach identifies and safeguard essential neurons while strategically introducing noise into non-essential neurons. Our experimental results demonstrate that our method successfully enhances both robustness and efficiency across diverse attack scenarios, model architectures, and datasets.

## 1 Introduction

Deep Neural Networks (DNNs) are at the forefront of technological advancements, powering a multitude of intelligent applications across various sectors (Al-Qizwini et al., 2017; Bai et al., 2018; Fujiyoshi et al., 2019). Yet, as DNNs become deeply integrated into mission-critical systems, two challenges emerge in DNN deployment. Firstly, when DNNs are used in vital decision-making tasks, their vulnerability to adversarial attacks becomes a serious concern. From a self-driving car misinterpreting a traffic sign due to subtle, maliciously introduced perturbations, to defense systems being deceived into false detection, the outcomes could be catastrophic (Carlini & Wagner, 2017; Goodfellow et al., 2014; Kurakin et al., 2016; Madry et al., 2017; Moosavi-Dezfooli et al., 2016). Secondly, deploying DNNs in resource-constrained environments relies on execution efficiency due to the increasing computational needs. The key challenge in this paper is to enhance robustness of DNNs when under adversarial attacks without compromising execution efficiency. Prior algorithmic methods for adversarial attacks often incur significant overheads, i.e., compromising performance, and without considering the implementation efficiency.

Recent studies indicate that the introduction of noise can improve both the robustness and efficiency of models, as evidenced by several research papers (Cohen et al., 2019; He et al., 2019; Xiao et al., 2020; Wu et al., 2020). Many techniques dedicated to improving the efficiency of neural networks can be treated as noise injection techniques, such as sparse noise (Fu et al., 2021b; Guo et al., 2018; Madaan et al., 2020; Sehwag et al., 2020; Ye et al., 2019; Gopalakrishnan et al., 2018; Gui et al., 2019), quantization noise (Fu et al., 2021a; Galloway et al., 2017; Lin et al., 2019), and approximation noise (Guesmi et al., 2021), have been employed to enhance both the resilience and computational efficiency of deep neural networks. Additionally, efforts have been made to design hardware accelerators specifically for this kind of algorithm (Fu et al., 2021c; Guesmi et al., 2021). However, these methods apply noise injections **uniformly** to all neurons, and such aggressive strategies inevitably compromise model accuracy.

We hypothesize that only a subset of neurons are essential for representation learning while the rest can tolerate noise perturbations without affecting overall accuracy. To address this, we introduce **non-uniform noise injection** to enhance DNN robustness. Instead of perturbing all neurons, our method protects identified essential neurons, bringing noise to only non-essential ones to enhance ro-

bustness. The key is to identify the essential neurons effectively and efficiently. With that, we adopt a learning-based approximate method to perform detection. Additionally, we propose to directly populate non-essential neurons with approximate values used for detection, as a way to reduce computation costs. We further investigate the noise injection granularity and propose structured noise injection to have more efficiency improvements. Thus, we present non-uniform noise injection to enhance both efficiency and robust accuracy. Our contributions are as follows:

- We are the first to identify that non-uniform noise injection is better than uniform one and advocate that when fine-grained design, noise injection has the potential to substantially improve the robustness of DNNs beyond their initial performance levels.
- We design a novel algorithm, which can efficiently select the essential and non-essential neurons and a non-uniform noise injection is brought via approximation to enhance DNN adversarial robustness while preserving clean accuracy. In addition, our method can be used orthogonally with adversarial training and also achieves better results than the original adversarial training methods.
- We further conduct the hardware performance analysis of our algorithm, yielding preliminary results that demonstrate its potential for efficiency.
- Using a variety of DNN models across different datasets and exposed to five distinct adversarial attacks with varying adversarial perturbations, we showcase that our algorithm consistently exhibits significantly higher robustness compared to the original model. When used in combination with adversarial training, it is also superior to the original adversarial training methods.

## 2 MOTIVATION

In this section, we analyze the limitations of existing robustness methods based on noise injection, and then present the assumptions of our non-uniform noise injection algorithm.

### 2.1 LIMITATIONS OF NOISE INJECTION-BASED METHODS

Model compression techniques, originally conceived to enhance execution efficiency, are now under the lens for their implications on adversarial robustness (Ye et al., 2019; Ahmad & Scheinkman, 2019). While techniques like weight pruning and quantization, when incorporated into adversarial training, have demonstrated boosted robust accuracy and reduced model sizes (Sehwag et al., 2020; Madaan et al., 2020; Fu et al., 2021b), they come with inherent limitations. For example, when model compression methods operate on model weights to force sparse connections or low precision, all neuron-wise activations become less effectual compared with activations of the dense counterpart(Lin et al., 2019; Zhuang et al., 2020).

Intuitively, model compression can be interpreted as using noise injection to boost model robustness. However, when model compression techniques impose such noise injections **uniformly across neurons**—be it due to pruned weights or reduced precision—there will be an inevitable dip in model accuracy. This way of uniform noise injection overlooks the nuanced contribution of individual neurons—leading to significant information loss and reduced clean accuracy. As illustrated in Figure 1, as the sparsity becomes more and more uniform, it can be seen that the clean accuracy of the model is rising, but the robustness is decreasing. Based on this observation, we propose a hypothesis: *only a subset of neurons plays a critical role in representation learning and retains high model accuracy*. The majority of neurons can endure targeted noise injection without adversely affecting the overall robust accuracy of DNNs.

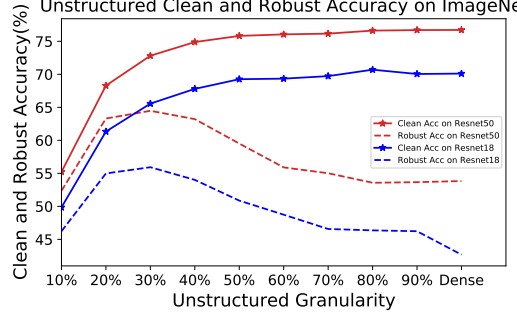

Figure 1: As the noise injection becomes more uniform, the trade-off between clean accuracy and robust accuracy. Injecting noise into fewer neurons achieves a small decrease in clean accuracy and a significant increase in robustness accuracy compared to the original model.

## 2.2 Non-uniform Noise Injection for Enhanced Robustness

We propose a **non-uniform noise injection** method via hardware-efficient approximation to enhance DNN adversarial robustness. Instead of uniform inject into all neurons, our data-dependent method can precisely identify and keep the critical neurons that are contributing more to model accuracy, while perturbing the remaining trivial neurons with approximate values to improve robust accuracy.

Then, we introduce **structured patterns** in noise injection granularity, ensuring that essential neurons remain intact as depicted in Figure 2. This approach promotes regular data access and execution. Our evaluations confirm that structured noise injections maintain robust accuracy comparable to fine-grained ones. Unlike the static sparsity in traditional weight pruning (Han et al., 2015; Niu et al., 2020; Roy et al., 2021; Zaheer et al., 2020), our method offers dynamic adaptability, perturbing activation values based on input samples. As demonstrated in Figure 2, using irregular 50% and structured 2:4 examples, we apply a Top-K selection to each activation matrix, preserving the highest K values and injecting noise into the rest. Furthermore, our structured pattern applied to the activation matrix enhances hardware efficiency. This method selects only the largest N elements within every M-element column. Viewing from a neuron perspective, essential neurons are retained, while others are perturbed, which will be further detailed in the subsequent section.

## 3 APPROACH

Building on the hypothesis that only a subset of neurons are critical for model inference, we can introduce noise injection to the non-essential neurons as a defense against adversarial attacks. Our challenge is to locate precisely these essential neurons. Drawing from approximation studies (Achlioptas, 2001; Ailon & Chazelle, 2009; Vu, 2016), we propose a learning-based method to identify essential neurons and inject noise into the non-essential neurons. Our approach not only retains model accuracy but also enhances adversarial robustness.

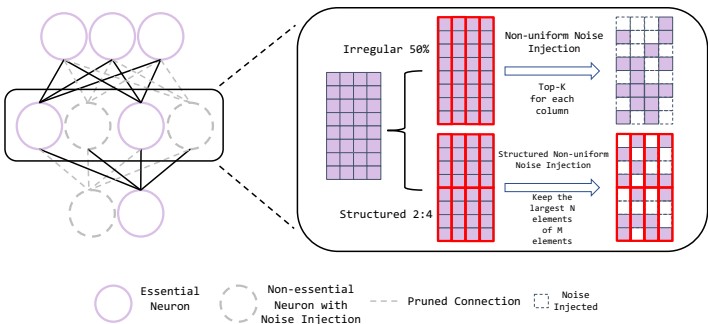

Figure 2: The concept of non-uniform noise injection from left: neuron perspective, where essential neurons are kept intact to contribute model accuracy and non-essential neurons are perturbed to improve robustness; and right: activation matrix perspective, where irregular noise injection on Top-K activations and structured noise injection keeps the largest N elements of M elements.

### 3.1 Learning-based Approximation Method

We introduce layer-wise approximation as $\tilde{z} = \tilde{W}Px + \tilde{b}$, where $\tilde{W} \in \mathbb{R}^{n \times k}$ and $\tilde{b} \in \mathbb{R}^n$ are trainable parameters, $P \in \sqrt{\frac{3}{k}} \cdot \{-1, 0, 1\}^{k \times d}$ is a sparse random projection matrix. Note that the approximate vector $\tilde{z}$ has the same dimension with the original output vector $z$. Since the approximate data is only used for the selection of essential neurons and injection of noise, we incorporate quantization to decrease the bit-width of approximation parameters to further reduce computation costs. Specifically, we apply a one-time quantization step on $\tilde{W}$ and $\tilde{b}$ to INT4 fixed-point arithmetic.

**Learning process of approximation parameters.** The trainable parameters, $\tilde{W}$ and $\tilde{b}$, are learned through minimizing the mean squared error (MSE) as the optimization target: $L_{MSE} = \frac{1}{B}||z - \tilde{z}||_2^2 = \frac{1}{B}||(Wx + b) - (\tilde{W}Px + \tilde{b})||_2^2$ where B is the mini-batch size. The random projection matrix $P$ is not trainable and stays constant after initialization.

## 3.2 SELECTION OF ESSENTIAL NEURONS AND NOISE INJECTION

After we obtain optimized approximation parameters, we can use the approximate results $\tilde{z}$ to estimate the importance of individual neurons and select those with higher magnitude among $\tilde{z}$. We can select essential neurons by comparing the approximate results with thresholds. A neuron is regarded as essential if its approximation from $\tilde{z}$ is larger than the predefined threshold. Specifically, we can generate a binary mask $m \in \{0,1\}^n$, work as a map of essential neurons to keep and nonessential neurons to inject noise, where $m_i$ equals 1 when the neurons are essential while it switches to 0 when the neurons are non-essential. In this way, we can estimate which neurons are essential without actual computations.

Once we identify the non-essential neurons, we can inject noise as a kind of perturbation. Instead of adding a noise term drawn from a normal or uniform distribution to the outputs, we propose to directly populate the non-essential neuron with approximate values drawn from $\tilde{z}$ that are computed in the selection step. Overall, the outputs of a NN layer under our method can be formulated as

$$z^{'} = z \odot m + \tilde{z} \odot (1 - m), \tag{1}$$

where the $\odot$ denotes point-wise multiplication.

## 3.3 PRESERVATION OF CLEAN ACCURACY

With the aforementioned algorithm, we can make the essential neurons maintain clean accuracy, while the non-essential neurons are injected with noise, which has a boosting effect on the adversarial robustness. In this section, we will provide the theoretical support for our approach.

We start with the proof that there is almost no clean accuracy drop by using our critical neuron selection algorithm. From another perspective, our essential neuron selection algorithm is an algorithm that dynamically prunes in a low-dimensional space. In other words, to prove that this algorithm has no loss in clean accuracy is to demonstrate that this transformation in low-dimensional space has almost no effect on the accuracy of matrix-matrix multiplication or matrix-vector multiplication. To simplify matters, we will focus solely on treating each operation within a sliding window of the convolution layer or the entirety of the fully connected (FC) layer, considering them as individual basic optimization problems for a single input sample. Each output activation $z_i$ is generated by the inner production:

$$z_i = \varphi \left( \langle x_i, W_j \rangle \right) \tag{2}$$

where $x_i$ is the $i$-th row in the matrix of input feature maps and for FC layer, there is only one $x$ vector. $W_j$ is the $j$-th column of the weight matrix $W$, and $\varphi(\cdot)$ is usually the activation function, here we omit the bias for simplicity. After defining Eq. 2 in this way, since matrix-matrix multiplication or matrix-vector multiplication consists of inner products, all we have to prove is that there exists a mapping of lower dimensional spaces that still gives a good approximation to inner products in higher dimensional spaces.

In dimensional transformations, according to the relationship between inner product and the Euclidean distance, the preservation of inner product is to preserve the Euclidean distance between two points. The following lemma has this effect.

**Lemma 1.** (Johnson, 1984). Given $0 < \epsilon < 1$, a set of $N$ points in $\mathbb{R}^d$ (i.e., all $x_i$ and $W_j$ ), and a number of $k > O\left(\frac{\log(N)}{\epsilon^2}\right)$, there exists a linear map $f : \mathbb{R}^d \Rightarrow \mathbb{R}^k$ such that $(1 - \epsilon) \|x_i - W_j\|^2 \le \|f(x_i) - f(W_j)\|^2 \le (1 + \epsilon) \|x_i - W_j\|^2$.

For any given $x_i$ and $W_j$ pair, where $\epsilon$ is a hyper-parameter to control the approximation error, i.e., larger $\epsilon \Rightarrow$ larger error. This lemma is a dimension-reduction lemma, named Johnson-Lindenstrauss Lemma(JLL)(Johnson, 1984), which states that a collection of points within a high-dimensional space can be transformed into a lower-dimensional space, where the Euclidean distances between these points remain closely preserved.

Random projection(Achlioptas, 2001; Ailon & Chazelle, 2009; Vu, 2016) has found extensive use in constructing linear maps $f(\cdot)$. In particular, the original $d$-dimensional vector is projected to a $k$-dimensional space, where $k \ll d$, utilizing a random $k \times d$ matrix $\mathbf{P}$. Consequently, we can reduce

the dimension of all $x_i$ and $W_j$ by applying this projection.

$$f(x_i) = \frac{1}{\sqrt{k}}\mathbf{P}x_i \in \mathbb{R}^k, \quad f(W_j) = \frac{1}{\sqrt{k}}\mathbf{P}W_j \in \mathbb{R}^k \tag{3}$$

The random projection matrix $\mathbf{P}$ can be generated from Gaussian distribution (Ailon & Chazelle, 2009). In this paper, we adopt a simplified version, termed as sparse random projection (Achlioptas, 2001; Bingham & Mannila, 2001; Li et al., 2006) with $Pr(\mathbf{P}_{pq} = \sqrt{s}) = \frac{1}{2s}; \quad Pr(\mathbf{P}_{pq} = 0) = 1 - \frac{1}{s}; \quad Pr(\mathbf{P}_{pq} = -\sqrt{s}) = \frac{1}{2s}$ for all elements in $\mathbf{P}$. This $\mathbf{P}$ only has ternary values that can remove the multiplications during projection, and the remaining additions are very sparse. Therefore, the projection overhead is negligible compared to other high-precision multiplication operations. Here we set $s = 3$ with 67% sparsity in statistics.

When $\epsilon$ in Lemma 1. is sufficiently small, a corollary derived from Johnson-Lindenstrauss Lemma (JLL) yields the following norm preservation:

**Corollary 1.** (Instructors Sham Kakade, 2009) For $\mathbf{Y} \in \mathbb{R}^d$. If the entries in $\mathbf{P} \subset \mathbb{R}^{k \times d}$ are sampled independently from $N(0, 1)$. Then,

$$Pr\left[(1-\epsilon)\|\mathbf{Y}\|^2 \le \|\frac{1}{\sqrt{k}}\mathbf{P}\mathbf{Y}\|^2 \le (1+\epsilon)\|\mathbf{Y}\|^2\right] \ge 1 - O\left(\epsilon^2\right). \tag{4}$$

where $\mathbf{Y}$ could be any $x_i$ or $W_j$. This implies that the preservation of the vector norm is achievable with a high probability, which is governed by the parameter $\epsilon$. Given these basics, we can further get the inner product preservation:

**Theorem 1.** Given a set of $N$ points in $\mathbb{R}^d$ (i.e. all $x_i$ and $W_j$), and a number of $k > O\left(\frac{\log(N)}{\epsilon^2}\right)$, there exists random projection matrix $\mathbf{P}$ and a $\epsilon_0 \in (0,1)$, for $0 < \epsilon \le \epsilon_0$ we have

$$Pr\left[\left|\left\langle \frac{1}{\sqrt{k}}\mathbf{P}x_i, \frac{1}{\sqrt{k}}\mathbf{P}W_j \right\rangle - \langle x_i, W_j \rangle\right| \le \epsilon\right] \ge 1 - O\left(\epsilon^2\right). \tag{5}$$

for all $x_i$ and $W_j$.

which indicates the low-dimensional inner product $\langle \frac{1}{\sqrt{k}}\mathbf{P}x_i, \frac{1}{\sqrt{k}}\mathbf{P}W_j \rangle$ can still approximate the original high-dimensional one $\langle x_i, W_j \rangle$ in Eq. 2 if the reduced dimension is sufficiently high. Therefore, it is possible to calculate Eq. 2 in a low-dimensional space for activation estimation and select the essential neurons. The detailed proof can be found in the Appendix A.

## 3.4 IMPROVEMENT OF ADVERSARIAL ROBUSTNESS

Regarding proof of robustness improvement, Pinot et al. (2019) has demonstrated that injecting noise into a deep neural network can enhance the model's resilience against adversarial attacks. A deep neural network can be considered as a probabilistic mapping $M$, which maps the input $\mathcal{X}$ to $\mathcal{Z}$ via $M : \mathcal{X} \to P(\mathcal{Z})$. According to Pinot et al. (2019), the risk optimization term of the model is defined as: $\text{Risk}(M) := \mathbb{E}_{(x,z)\sim\mathcal{D}}\left[\mathbb{E}_{z'\sim M(x)}\left[\mathbb{1}\left(z' \ne z\right)\right]\right]$ In the adversarial attack scenario, the model risk optimization term becomes the: $\text{Risk}_\alpha(M) := \mathbb{E}_{(x,z)\sim\mathcal{D}}\left[\sup_{\|\tau\|_\mathcal{X}\le\alpha} \mathbb{E}_{z'\sim M(x+\tau)}\left[\mathbb{1}\left(z' \ne z\right)\right]\right]$ where $\tau$ is the adversarial perturbation applied to the input sample, $\alpha$ is treated as the upper limit of perturbation. After obtaining these, Pinot et al. (2019), (Theorem 1) proved that noise sampled from the Exponential Family can ensure robustness. Finally, the robustness of the neural network with noise injection can be expressed by the following theorem:

**Theorem 2.** (Pinot et al., 2019) Let $M$ be the probabilistic mapping at hand. Let us suppose that $M$ is robust, then:

$$|\text{Risk}_\alpha(M) - \text{Risk}(M)| \le 1 - e^{-\theta}\mathbb{E}_x\left[e^{-H(M(x))}\right] \tag{6}$$

where $H$ is the Shannon entropy $H(p) = -\sum_i p_i \log(p_i)$.

This theorem provides a means of controlling the accuracy degradation when under attack, with respect to both the robustness parameter $\theta$ and the entropy of the predictor. Intuitively, as the injection of noise increases, the output distribution tends towards a uniform distribution for any input.

Consequently, as $\theta \to 0$ and the entropy $H(M(x)) \to \log(K)$, $K$ is the number of classes in the classification problem, both the risk and the adversarial risk tend towards $1/K$. Conversely, when no noise is introduced, the output distribution for any input resembles a Dirac distribution. In this scenario, if the prediction for an adversarial example differs from that of a regular one, $\theta \to \infty$ and $H(M(x)) \to 0$. Therefore, the design of noise needs to strike a balance between preserving accuracy and enhancing robustness against adversarial attacks, which proves our motivation.

### 3.5    Implications for Efficient Execution

As indicated in Eq. 1, the computed pattern is a mixture of precise computation and approximate computation in the form of non-uniform noise perturbation. For precise computation, only essential neurons need to be computed and non-essential neurons are from approximation. Hence, we can skip precise computations of non-essential neurons, leading to potential performance speedup and energy saving in the similar spirit of accelerated execution of activation sparsity. In addition, our approximate method incurs a small amount of low-precision operations.

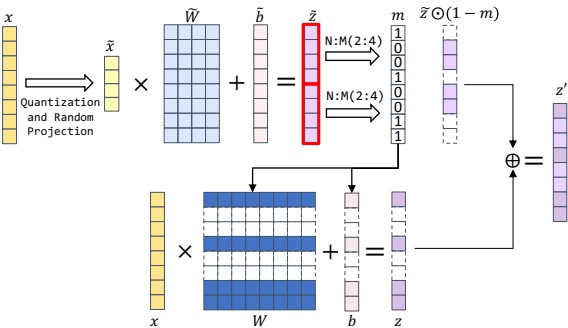

Figure 3: Structured Non-uniform Noise Injection: After getting the approximate module from fine-tuning, the Top-K algorithm is utilized to take out the index of the largest value of $N$, and then the corresponding mask $m$ is generated. Accurate module carries out the N:M Sparsity through the mask $m$, and the final result is still a mixture of approximate and accurate modules.

**On noise injection granularity.** While non-uniform noise injection can improve execution efficiency from computation skipping of essential neurons, the unconstrained and unstructured noise injection patterns would increase the hardware design complexity and execution overheads from irregular data access and low data reuse.

Taking inspiration from the sparsity-oriented designs, we raise the hypothesis that noise injection granularity can be constrained in a similar way as sparsity constraints. In particular, structured sparsity, denoted as N:M, is an emerging trend that preserves N elements in every 1×M vector of a dense matrix. This fine-grained approach offers more combinations than block sparsity. An example is the 1:2 and 2:4 structured pruning techniques for neural network weight introduced in NVIDIA Ampere. Such techniques aim for efficiency and faster inference without sacrificing performance.

As shown in Figure 3, our revised method, i.e., structured noise injection, chooses N essential neurons out of a vector of M neurons and injects noise into the rest. Note that here we perform noise injection dynamically on neurons or activations, not static weight sparsity, despite the similarity in granularity.

## 4    Experiments

In this section, we mainly present the following advantages of our non-uniform noise injection method. Our detailed evaluation methodology is in Appendix B. First, non-uniform noise injection can enhance adversarial robustness of the model while retaining clean accuracy. Second, our algorithm can be combined with adversarial training techniques to advance robustness further. Third, we experiment with the algorithm's efficiency gains and initially explore hardware feasibility.

In our first experiment, we employ the ResNet18 and ResNet50 CNN architectures(He et al., 2016) that are trained on the CIFAR-10 dataset (Krizhevsky et al., 2009) and the ImageNet-2012 dataset (Russakovsky et al., 2015). Non-uniform noise injection is applied only to the convolutional layers, and we implemented these models using the PyTorch framework (Paszke et al., 2019). To verify that our non-uniform noise injection can be trained under adversarial conditions, we selected the structured 4:8 pattern, which demonstrates the best robustness for ResNet18 and ResNet50 on CIFAR-10. We apply adversarial training using two methods, FGSM-RS (Wong et al., 2020) and PGD-10 (Madry et al., 2017). To assess the algorithm's granularity stability, we conduct experiments with larger granularity, as described in Appendix C.

## 4.1 DEFEND WITH NON-UNIFORM NOISE INJECTION

As shown in Table 1, our method has almost no loss in clean accuracy and exhibits better robustness than the original models. Specifically, the best robustness gains are both on structured 4:8, 10.49% on Resnet18 under BIM attack, and 6.61% on Resnet50 with DeepFool attack, respectively. The second best is the Irregular 50%, and all other noise injection methods have better robustness than the original model.

Table 1: The clean accuracy(%) and robust accuracy(%) of our non-uniform noise injection algorithm of the ResNet18 and ResNet50 on CIFAR-10 under various adversarial attack methods with different perturbation $\epsilon$.

| | | ResNet18 | | | | | | ResNet50 | | | | |
|---|---|---|---|---|---|---|---|---|---|---|---|---|
| Granularity | Clean | PGD-40 | FGSM | CW | DeepFool | BIM | Clean | PGD-40 | FGSM | CW | DeepFool | BIM |
| | | $\epsilon = 4/255$ | $\epsilon = 4/255$ | $\epsilon = 0.5$ | $\epsilon = 4/255$ | $\epsilon = 4/255$ | | $\epsilon = 4/255$ | $\epsilon = 4/255$ | $\epsilon = 0.5$ | $\epsilon = 4/255$ | $\epsilon = 4/255$ |
| Original | 91.41 | 7.50 | 46.61 | 12.34 | 32.32 | 16.68 | 92.49 | 4.03 | 34.61 | 11.66 | 27.46 | 7.76 |
| Irregular 50% | 90.50 | 12.70 | 50.07 | 16.30 | 36.43 | 23.49 | 91.41 | 6.07 | 38.97 | 16.48 | 32.68 | 11.54 |
| Irregular 80% | 91.32 | 8.12 | 47.49 | 15.29 | 33.67 | 17.74 | 91.42 | 4.71 | 37.55 | 15.23 | 30.38 | 9.15 |
| Irregular 90% | 91.96 | 8.18 | 47.20 | 13.65 | 33.47 | 17.49 | 92.52 | 4.77 | 37.36 | 13.75 | 30.54 | 9.10 |
| Structured 4:8 | 90.40 | **15.68** | **52.53** | **19.78** | **39.77** | **27.17** | 91.06 | **6.67** | **39.70** | **17.03** | **34.07** | **12.19** |
| Structured 5:8 | 90.98 | 11.37 | 50.12 | 16.17 | 37.09 | 22.52 | 91.22 | 5.29 | 37.88 | 14.84 | 31.50 | 10.34 |
| Structured 6:8 | 91.72 | 9.17 | 48.43 | 14.37 | 34.83 | 19.61 | 91.41 | 4.83 | 37.21 | 13.40 | 30.79 | 9.32 |

Table 2: The The clean accuracy(%) and robust accuracy(%) of our non-uniform noise injection algorithm of the ResNet18 and ResNet50 on ImageNet-2012 under various adversarial attack methods with different perturbation $\epsilon$.

| | | ResNet18 | | | | | | ResNet50 | | | | |
|---|---|---|---|---|---|---|---|---|---|---|---|---|
| Granularity | Clean | PGD-40 | FGSM | CW | DeepFool | BIM | Clean | PGD-40 | FGSM | CW | DeepFool | BIM |
| | | $\epsilon = 0.0005$ | $\epsilon = 0.0005$ | $\epsilon = 0.2$ | $\epsilon = 0.0005$ | $\epsilon = 0.0005$ | | $\epsilon = 0.0005$ | $\epsilon = 0.0005$ | $\epsilon = 0.2$ | $\epsilon = 0.0005$ | $\epsilon = 0.0005$ |
| Original | 70.10 | 42.05 | 42.69 | 42.78 | 39.31 | 38.50 | 76.64 | 51.26 | 53.83 | 55.05 | 50.06 | 47.11 |
| Irregular 50% | 69.26 | 50.78 | 50.87 | 51.42 | 48.54 | 47.85 | 75.83 | 58.89 | 59.52 | 60.95 | 58.25 | 55.65 |
| Irregular 80% | 70.71 | 45.66 | 46.37 | 46.42 | 42.96 | 42.10 | 76.63 | 50.73 | 53.57 | 54.90 | 49.89 | 46.57 |
| Irregular 90% | 70.06 | 45.48 | 46.23 | 46.18 | 42.71 | 41.86 | 76.70 | 51.00 | 53.67 | 55.11 | 49.94 | 46.92 |
| Structured 4:8 | 68.98 | **53.44** | **53.50** | **53.67** | **51.25** | **50.80** | 75.07 | **62.49** | **62.73** | **63.68** | **61.36** | **59.99** |
| Structured 5:8 | 69.66 | 50.03 | 50.26 | 50.37 | 47.47 | 46.92 | 75.79 | 58.52 | 59.26 | 60.32 | 57.35 | 55.30 |
| Structured 6:8 | 69.96 | 47.85 | 48.21 | 48.41 | 45.13 | 44.43 | 76.13 | 55.09 | 56.51 | 57.67 | 53.97 | 51.44 |

When looking at Table 2, the results obtained from ImageNet-2012 align closely with those from CIFAR-10, demonstrating the scalability of our non-uniform noise injection approach to more complex tasks. Notably, the structured 4:8 method outperforms others, achieving a 10.81%∼12.30% and 8.63%∼12.88% robustness increase over the original ResNet18 and ResNet50 networks, respectively. The Irregular 50% approach also delivers robust results, enhancing accuracy by 8.18% to 9.35% for ResNet18 and between 5.69% to 8.54% for ResNet50 across various attack scenarios. Other granularity methods contribute accuracy improvements ranging from 3% to 6%, with exceptions being Irregular 80% and Irregular 90%, which see minor robustness losses under Deepfool and BIM attacks. Importantly, in terms of clean accuracy, our methods produce negligible losses.

These outcomes demonstrate the efficacy of non-uniform noise injection in enhancing adversarial robustness. It retains the original model accuracy on clean data, suggesting that essential neurons experience minimal noise injection, and elevates network robustness, indicating that noise injection to non-essential neurons contributes to this improvement.

## 4.2 JOINT WITH ADVERSARIAL TRAINING

As illustrated in Table 3, we can observe that **(1)** our non-uniform noise injection consistently improves robust accuracy across various PGD attack settings, for all networks and adversarial training

methods. Specifically, when applying FGSM-RS, our algorithm achieves a 2.95%/3.19% higher robust accuracy under PGD-20 attacks on ResNet18 and ResNet50, respectively. Furthermore, when integrated with PGD-10 training,we observe improvements of 2.08% and 1.89% on ResNet18 and ResNet50, respectively. Also, our method maintains a consistent effect as the strength of the attack increases. **(2)** Our approach has a negligible drop in clean accuracy. In particular, when applying FGSM-RS and PGD-10, resnet18 shows a decrease in clean accuracy of only 1.86% and 0.46%, respectively. For ResNet50, the drop in both methods is only 0.81% and 0.86%.

We further compare our method with the quantization uniform noise injection method, see Appendix D for details. This reaffirms our hypothesis that essential neurons without any noise injections maintain clean accuracy and that non-essential neurons injected with noise can bring about an enhancement of robustness.

Table 3: The clean accuracy(%) and robust accuracy(%) of our non-uniform noise injection structured 4:8 algorithm joint with adversarial training of the ResNet18 and ResNet50 on CIFAR-10 under different adversarial attack methods with perturbation $\epsilon = 8/255$.

| Adversarial Training Method | ResNet18 | | | | ResNet50 | | | |
|---|---|---|---|---|---|---|---|---|
| | Clean | PGD-20 | PGD-40 | PGD-100 | Clean | PGD-20 | PGD-40 | PGD-100 |
| | | $\epsilon = 8/255$ | $\epsilon = 8/255$ | $\epsilon = 8/255$ | | $\epsilon = 8/255$ | $\epsilon = 8/255$ | $\epsilon = 8/255$ |
| Original | 91.41 | 1.45 | 0.78 | 0.61 | 92.49 | 0.09 | 0.04 | 0.02 |
| Structured 4:8 | 90.40 | **5.92** | **3.85** | **2.50** | 91.06 | **0.17** | **0.19** | **0.03** |
| FGSM-RS | 84.77 | 34.51 | 33.71 | 33.32 | 86.08 | 38.09 | 37.28 | 36.26 |
| FGSM-RS+Structural 4:8 | 82.91 | **37.46** | **36.67** | **35.25** | 85.27 | **41.28** | **41.10** | **39.47** |
| PGD-10 | 81.08 | 43.89 | 43.52 | 43.19 | 83.96 | 46.43 | 46.21 | 44.96 |
| PGD-10+Structural 4 :8 | 80.62 | **45.97** | **45.06** | **43.25** | 83.10 | **48.32** | **47.59** | **45.38** |

## 4.3 HARDWARE EFFICIENCY

By matching the hardware capabilities to the algorithm requirements, we can achieve better hardware efficiency. A major factor is using relatively lower cost of low-precision arithmetic vs. high-precision. Because multiplication characterizes each bit against each other, scaling from 4- to 16-bits increases the intensity by 16x, not 4x. As a result, the cost of approximation is low compared to full-precision calculation.

In our analysis, because the hardware uses a precision-scalable Matrix-Vector Unit (MVU), we convert all operations to BitOps, which approximate the bit-level intensity of operations. For reference, sixteen BitOps = four 4-bit additions = one 4-bit multiplication. As it pertains to performance, Figure 4 shows that the noise injection granularity affects on various statistics. This figure highlights two key findings: 1) The effect on noise injection alone and 2) The effect of structured version.

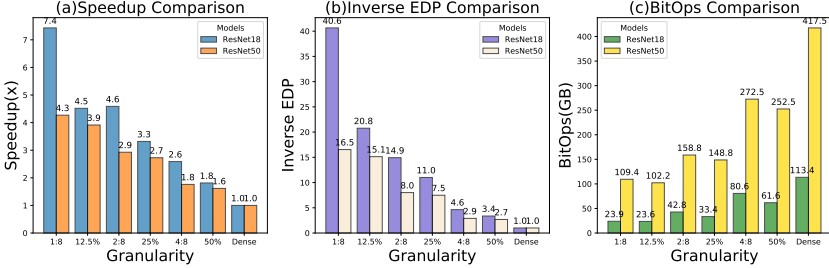

Figure 4: Normalized value of various metrics across different levels of perturbation generated from our hardware simulator. NOTE: sixteen BitOps = four 4-bit additions = one 4-bit multiplication.

First, with the relative intensity of full-precision calculation, increasing perturbation alone effectively improves speed and efficiency. This is because, as mentioned prior, *high precision execution dominates performance*. A reduction in the high precision stage therefore has a significant effect on performance.

Second, when using a structured perturbation scheme, speed and efficiency are improved relative to an equivalent unstructured scheme. This is because with constrained, predictable execution patterns,

we can expand the MVU without significant overhead, allowing for a higher throughput and faster execution despite a worse mapping efficiency. Additionally, with structured scheme, there is more data reuse, reducing the SRAM energy. From these two points, we can see structured dynamic perturbation is an effective means to vastly improve inference performance.

## 5 RELATED WORK

**Adversarial Attacks & Defense** Adversarial attacks, as discussed in Croce & Hein (2020), pose a significant threat to the deployment of machine learning (ML) models. These attacks involve subtly altering input data to deceive ML models, leading to incorrect predictions. The concept of "adversarial robustness" measures a neural network's ability to resist such attacks by comparing its accuracy on clean data to its accuracy on adversarially perturbed data (Carlini & Wagner, 2017). Without protective measures, ML models can experience a significant drop in accuracy, often exceeding 20%, even under basic attacks. To defend against adversarial attacks, various methods have been proposed, including defensive algorithms (Cohen et al., 2019; Dhillon et al., 2018; He et al., 2019; Jeddi et al., 2020). These methods aim to enhance a model's resistance to attacks but may lead to performance trade-offs when dedicated hardware support is unavailable. Adversarial training (Shafahi et al., 2019; Szegedy et al., 2013; Wong et al., 2020; Zhang et al., 2019a) is a notable approach where models are trained using both regular and adversarial samples to expose and mitigate vulnerabilities. However, this method often sacrifices performance on clean data(Tsipras et al., 2018; Zhang et al., 2019b; Nakkiran, 2019; Stutz et al., 2019) and requires intensive re-training (Wang et al., 2020). Moreover, it lacks the flexibility to dynamically adjust the trade-off between clean and robust accuracy. It's important to note that while adversarial training can improve model resilience, it doesn't necessarily optimize computational efficiency. This highlights the need to explore alternative techniques that reduce computational costs without relying on specialized hardware.

**Robustness and Efficiency** To make deep neural networks (DNNs) work better, researchers have looked into three main ways: making them smaller, and resilient to attacks. For instance, in the sparsity and robustness, Guo et al. (2018) examines how introducing sparsity in DNN architectures can increase their resilience against adversarial attacks. More comprehensively, Ye et al. (2019) explores various techniques like adversarial training, robust regularization, and model compression methods such as pruning, quantization, and knowledge distillation to enhance adversarial robustness. Some work, inspired by the lottery hypothesis(Frankle & Carbin, 2018) and adversarial training (Madry et al., 2017; Shafahi et al., 2019; Wong et al., 2020), combines pruning or sparse masking with adversarial training to obtain a sparse subnetwork with robustness (Sehwag et al., 2020; Madaan et al., 2020; Fu et al., 2021b). Our proposed algorithm in this paper falls under dynamic sparsity, incorporating non-uniform perturbations for finer-grained robustness exploration. Another research direction focuses on the robustness of low-precision or quantized DNNs. For example, Galloway et al. (2017) introduces robust binary neural networks that demonstrate increased adversarial robustness compared to full-precision networks. Defensive Quantization (Lin et al., 2019) controls the network's Lipschitz constant during quantization to maintain non-expansive adversarial noise during inference, striking a balance between efficiency and robustness. Additionally, Fu et al. (2021c) and its corresponding algorithm design (Fu et al., 2021a) explores the challenges of transferring between different quantization bits and introduces an algorithm for randomized accuracy switching during adversarial training and attack phases to improve robustness and efficiency. Another approach involves introducing noise through approximation computation, with defensive approximation ((Guesmi et al., 2021)) being a representative example. However, these methods usually introduce noises to all neurons, which makes it challenging for neurons to maintain clean accuracy.

## 6 CONCLUSION

In this work, we introduce a novel approach: non-uniform noise injection, which bridges the gap between adversarial robustness and execution efficiency. Inspired by the trade-off between robustness and clean accuracy, our data-dependent method can precisely identify and keep the critical neurons that are contributing more to model accuracy, while injecting noise to the remaining trivial neurons with approximate values to improve robust accuracy. We believe that our work sheds light on the importance of fine-grained noise injection in bolstering adversarial robustness, contributing to a deeper understanding of this critical aspect in the field of machine learning.

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

## A PROOF OF THE ALGORITHM FOR INNER PRODUCT PRESERVATION

**Theorem 1.** Given a set of $N$ points in $\mathbb{R}^d$ (i.e. all $x_i$ and $W_j$ ), and a number of $k > O\left(\frac{\log(N)}{\epsilon^2}\right)$, there exists random projection matrix $\mathbf{P}$ and a $\epsilon_0 \in (0,1)$, for $0 < \epsilon \leq \epsilon_0$ we have

$$Pr\left[\left|\left\langle \frac{1}{\sqrt{k}}\mathbf{P}x_i, \frac{1}{\sqrt{k}}\mathbf{P}W_j \right\rangle - \langle x_i, W_j\rangle\right| \leq \epsilon\right] \geq 1 - O\left(\epsilon^2\right).$$

for all $x_i$ and $W_j$.

**Proof.** According to the definition of inner product and vector norm, any two vectors $a$ and $b$ satisfy

$$\begin{cases} \langle \mathbf{a}, \mathbf{b}\rangle = \left(\|\mathbf{a}\|^2 + \|\mathbf{b}\|^2 - \|\mathbf{a} - \mathbf{b}\|^2\right)/2 \\ \langle \mathbf{a}, \mathbf{b}\rangle = \left(\|\mathbf{a} + \mathbf{b}\|^2 - \|\mathbf{a}\|^2 - \|\mathbf{b}\|^2\right)/2 \end{cases}. \tag{7}$$

It is easy to further get

$$\langle \mathbf{a}, \mathbf{b}\rangle = \left(\|\mathbf{a} + \mathbf{b}\|^2 - \|\mathbf{a} - \mathbf{b}\|^2\right)/4. \tag{8}$$

Therefore, we can transform the target in Eq. 2 to

$$|\langle f(x_i), f(W_j)\rangle - \langle x_i, W_j\rangle|$$
$$= \left|\|f(x_i) + f(W_j)\|^2 - \|f(x_i) - f(W_j)\|^2 - \|x_i + W_j\|^2 + \|x_i - W_j\|^2\right|/4 \tag{9}$$
$$\leq \left|\|f(x_i) + f(W_j)\|^2 - \|x_i + W_j\|^2\right|/4 + \left|\|f(x_i) - f(W_j)\|^2 - \|x_i - W_j\|^2\right|/4$$

which is also based on the fact that $|u - v| \leq |u| + |v|$. Now recall the definition of random projection in Eq. 3 of the main text

$$f(x_i) = \frac{1}{\sqrt{k}}\mathbf{P}x_i \in \mathbb{R}^k, \quad f(W_j) = \frac{1}{\sqrt{k}}\mathbf{P}W_j \in \mathbb{R}^k.$$

Substituting Eq. 3 into Eq. 9, we have

$$|\langle f(x_i), f(W_j)\rangle - \langle x_i, W_j\rangle|$$
$$\leq \left|\left\|\frac{1}{\sqrt{k}}\mathbf{P}x_i + \frac{1}{\sqrt{k}}\mathbf{P}W_j\right\|^2 - \|x_i + W_j\|^2\right|/4 + \left|\left\|\frac{1}{\sqrt{k}}\mathbf{P}x_i - \frac{1}{\sqrt{k}}\mathbf{P}W_j\right\|^2 - \|x_i - W_j\|^2\right|/4$$
$$= \left|\left\|\frac{1}{\sqrt{k}}\mathbf{P}(x_i + W_j)\right\|^2 - \|x_i + W_j\|^2\right|/4 + \left|\left\|\frac{1}{\sqrt{k}}\mathbf{P}(x_i - W_j)\right\|^2 - \|x_i - W_j\|^2\right|/4$$
$$\tag{10}$$

Further recalling the norm preservation in Eq. 4 of the main text: there exists a linear map $f: \mathbb{R}^d \Rightarrow \mathbb{R}^k$ and a $\epsilon_0 \in (0,1)$, for $0 < \epsilon \leq \epsilon_0$ we have

$$Pr\left[(1-\epsilon)\|\mathbf{Y}\|^2 \leq \|\frac{1}{\sqrt{k}}\mathbf{P}\mathbf{Y}\|^2 \leq (1+\epsilon)\|\mathbf{Y}\|^2\right] \geq 1 - O\left(\epsilon^2\right).$$

Substituting the Eq. 4 into Eq. 10 yields

$$P\left[\left|\left\|\frac{1}{\sqrt{k}}\mathbf{P}(x_i + W_j)\right\|^2 - \|x_i + W_j\|^2\right|/4 + \left|\left\|\frac{1}{\sqrt{k}}\mathbf{P}(x_i - W_j)\right\|^2 - \|x_i - W_j\|^2\right|/4 \ldots\right.$$
$$\left.\leq \frac{\epsilon}{4}\left(\|x_i + W_j\|^2 + \|x_i - W_j\|^2\right) = \frac{\epsilon}{2}\left(\|x_i\|^2 + \|W_j\|^2\right)\right] \ldots$$
$$\geq Pr\left(\left|\left\|\frac{1}{\sqrt{k}}\mathbf{P}(x_i + W_j)\right\|^2 - \|x_i + W_j\|^2\right|/4 \leq \frac{\epsilon}{4}\|x_i + W_j\|^2\right) \ldots \tag{11}$$
$$\times Pr\left(\left|\left\|\frac{1}{\sqrt{k}}\mathbf{P}(x_i - W_j)\right\|^2 - \|x_i - W_j\|^2\right|/4 \leq \frac{\epsilon}{4}\|x_i - W_j\|^2\right) \ldots$$
$$\geq \left[1 - O\left(\epsilon^2\right)\right] \cdot \left[1 - O\left(\epsilon^2\right)\right] = 1 - O\left(\epsilon^2\right).$$

Combining equation 9 and 11, finally we have

$$Pr\left[\left|\left\langle \frac{1}{\sqrt{k}}\mathbf{P}x_i, \frac{1}{\sqrt{k}}\mathbf{P}W_j \right\rangle - \langle x_i, W_j\rangle\right| \leq \frac{\epsilon}{2}\left(\|x_i\|^2 + \|W_j\|^2\right)\right] \geq 1 - O\left(\epsilon^2\right) \tag{12}$$

## B  DETAILED EXPERIMENTAL SETUP

**Hardware Experimental Setup.** We tested the effectiveness of our algorithm on a co-designed hardware implementation via an in-house cycle-accurate simulator. The simulator tracks key performance metrics and maps them to specific power values. SRAM power/area is estimated using CACTI, and all other components are synthesized using Synopsys Design Compiler on the FreePDK 45nm PDK.

To get an accurate reference for an optimized accelerator, we design our own hardware. The general hierarchy is simple, with a central Matrix-Vector Unit (MVU), core-specific SRAM, and registers. The MVU is purpose-designed to match the algorithm requirements as efficiently as possible. It which uses 4-bit multipliers as a precision-scaling element, offering both 4- and 16-bit multiplication while limiting additional hardware.

**Training Strategies.** For the first experiment, on CIFAR-10, we train the models using the momentum SGD optimizer for 200 epochs and fine-tuned them for 50 epochs. On ImageNet-2012, we used the Adam optimizer for training, running for 50 epochs initially and fine-tuning for an additional 10 epochs to achieve various noise injection ratios and structured sparsity. Our experiments with non-uniform noise injection involved three different noise injection ratios: 50%, 80%, and 90%. We also explore structured non-uniform noise injection with patterns of 4:8, 5:8, and 6:8.

For the second experiment, we follow the original papers' hyperparameter settings for adversarial training, which included a step size of $1.25\epsilon$ for FGSM-RS and 2 for PGD-7 training. During training, we fine-tune the models for 20 epochs with a batch size of 256, utilizing the SGD optimizer with a momentum of 0.9 and an initial learning rate of 0.02, along with a cyclic scheduler.

**Attacks.** To evaluate adversarial robustness, we deploy five attack methods on the first experiment: PGD (Madry et al., 2017), FGSM(Goodfellow et al., 2014), C&W (Carlini & Wagner, 2017), Deep-Fool (Moosavi-Dezfooli et al., 2016), and BIM (Kurakin et al., 2016). On CIFAR-10, C&W attack uses $\epsilon = 0.5$, and all other methods use $\epsilon = 4/255$, among which PGD-40 uses the step size of 1/255. These attacks are implemented using the Foolbox library (Rauber et al., 2017; 2020), adhering to its default settings. On ImageNet-2012, Except for the C&W attack, which employs an $l_2$ perturbation limit of $\epsilon = 0.2$, all other methods use $l_\infty$ attacks with a perturbation ceiling of $\epsilon = 0.0005$. We mainly focus on PGD-20, PGD-40 and PGD-100 (Madry et al., 2017) attacks with perturbation strengths $\epsilon = 8/255$, the step size is 2/255 in adversarial training.

**Additional Analysis.** Structured perturbation does not reduce the number of operations that need to occur when compared to unstructured perturbation, however, it constrains the execution so that key optimizations can be made. With unstructured, there is no guarantee that a weight will be reused or not. This makes it very difficult to scale up performance without also scaling the weight read bandwidth or vastly complicating control. With structured, however, there is now a guarantee that within a given window (We use 8 rows), there is a strong spatial locality across columns. With this avenue now open, we design to parallelize columns. We now load more activations at once, increased the lane width of the MVU, and left SRAM bandwidths the same. We see significant strides in performance from these changes. Figure 4 compares the speedup, efficiency, and operational intensity between structured and unstructured. We use EDP (Energy Delay Product) as a metric for efficiency. We first see that the enhanced parallelism allows for a greater reduction in execution time. Second, we see an overall improvement in execution efficiency. Lower execution time, while desirable on its own, decreases static energy consumption, which is especially important on chips with large amounts of hardware. Spatial locality enables reuse, which leads to fewer overall SRAM accesses and thus dynamic power. The combination of these leads to an approximately equivalent energy per inference compared to a comparable unstructured scheme. When factoring in the previous speedup, we see that the overall EDP is reduced when going from unstructured to an equivalent structured perturbation scheme.

A nuance to structured execution is that the optimal execution pattern is not always met. As seen by subplot C, structured can add some redundant operations due to poor boundary alignment. As such, VPUs will not be entirely full, leading to some redundant operations. This can be avoided with some additional hardware, but that itself could outweigh the cost of occasionally reduced efficiency. As such, we perform our analysis with no such hardware. Regardless, we see that it does not significantly effect performance, as it accounts for a lower overhead relative to the savings.

## C   ROBUSTNESS EXPERIMENTS AT LARGER GRANULARITY

**Experimental Setups.**  For the verification of the stability of hardware-friendly granularity, we continue with a series of experiments at a larger granularity in the case of a 50% noise injection, and we experiment on the performance of ResNet18 and ResNet50 on CIFAR-10 and ImageNet, with all the attack parameters the same as Tables 1 and 2 respectively.

**Results.**  As can be seen in Tables 4 and 5, our non-uniform noise injection is stable for different hardware-friendly structured sparsity, which means that our algorithm can be scaled to the appropriate hardware-friendly structured sparsity granularity as required.

Table 4: The clean accuracy(%) and robust accuracy(%) of larger granularity in the case of a 50% noise injection of the ResNet18 and ResNet50 on CIFAR-10 under various adversarial attack methods with different perturbation $\epsilon$.

| Granularity | Clean | PGD-40 | FGSM | CW | DeepFool | BIM |
|---|---|---|---|---|---|---|
| | | $\epsilon = 4/255$ | $\epsilon = 4/255$ | $\epsilon = 0.5$ | $\epsilon = 4/255$ | $\epsilon = 4/255$ |
| ResNet18 | 91.41 | 7.5 | 46.61 | 12.34 | 32.32 | 16.68 |
| Structural 8:16 | 91.25 | 16.86 | 55.42 | 21.54 | 42.31 | 29.35 |
| Structural 16:32 | 91.05 | 15.79 | 55.19 | 20.72 | 41.62 | 28.47 |
| Structural 32:64 | 91.01 | 14.97 | 55.06 | 20.42 | 41.48 | 27.83 |
| ResNet50 | 92.49 | 4.03 | 34.61 | 11.66 | 27.46 | 7.76 |
| Structural 8:16 | 92.12 | 7.40 | 42.44 | 18.03 | 35.96 | 13.57 |
| Structural 16:32 | 92.33 | 6.99 | 42.09 | 17.31 | 35.57 | 13.10 |
| Structural 32:64 | 92.26 | 7.42 | 42.97 | 17.77 | 36.03 | 13.46 |

Table 5: The clean accuracy(%) and robust accuracy(%) of larger granularity in the case of a 50% noise injection of the ResNet18 and ResNet50 on ImageNet under various adversarial attack methods with different perturbation $\epsilon$.

| Granularity | Clean | PGD-40 | FGSM | CW | DeepFool | BIM |
|---|---|---|---|---|---|---|
| | | $\epsilon = 0.0005$ | $\epsilon = 0.0005$ | $\epsilon = 0.2$ | $\epsilon = 0.0005$ | $\epsilon = 0.0005$ |
| ResNet18 | 70.10 | 42.05 | 42.69 | 42.78 | 39.31 | 38.50 |
| Structural 8:16 | 68.15 | 52.14 | 53.15 | 53.31 | 50.97 | 50.76 |
| Structural 16:32 | 68.18 | 51.76 | 52.73 | 52.88 | 50.49 | 50.27 |
| Structural 32:64 | 68.36 | 51.53 | 52.56 | 52.75 | 50.27 | 50.03 |
| ResNet50 | 76.64 | 51.26 | 53.83 | 55.05 | 50.06 | 47.11 |
| Structural 8:16 | 74.48 | 61.06 | 61.90 | 62.79 | 60.55 | 59.30 |
| Structural 16:32 | 74.57 | 60.51 | 61.27 | 62.02 | 59.82 | 58.50 |
| Structural 32:64 | 74.66 | 60.09 | 60.89 | 61.78 | 59.36 | 58.11 |

## D   COMPARISION WITH UNIFORM QUANTIZATION NOISE INJECTION

**Experimental Setups.**  To validate our motivation, we conduct an experimental comparison with quantization uniform noise injection. We use the same experimental setup for adversarial training as Table 3, with INT8, INT4, and INT2 chosen for the quantization method.

**Results.**  As shown in Table 6, under the adversarial training approach of FGSM, both INT8 and INT4 have a slight improvement in robust accuracy compared to the original model. However, when it comes to clean accuracy, quantization noise injection brings about a substantial decrease in clean accuracy. As the quantization bit decreases, there is a large drop in both clean accuracy and robust accuracy due to the error amplification effect(Lin et al., 2019). Under the adversarial training mode of PGD-10, quantization uniform noise injection begins to become unstable. Compared to the original model, both clean accuracy and robust accuracy fall dramatically.

Table 6: The clean accuracy(%) and robust accuracy(%) of quantization noise injection in the case of structured 4:8 noise injection of the ResNet18 on CIFAR10 under PGD-20 with perturbation $\epsilon = 8/255$ and step size is $2/255$.

| Adversarial Training Method | Clean | PGD-20 |
| --- | --- | --- |
| | | $\epsilon = 8/255$ |
| Original | 91.41 | 1.45 |
| Structured 4:8 | 90.40 | 5.92 |
| FGSM-RS | 84.77 | 34.51 |
| FGSM-RS+Structural 4:8 | 82.91 | 37.46 |
| FGSM-RS+INT 8 | 69.72 | 36.70 |
| FGSM-RS+INT 4 | 64.63 | 35.19 |
| FGSM-RS+INT 2 | 58.87 | 28.42 |
| PGD-10 | 81.08 | 43.89 |
| PGD-10+Structural 4 :8 | 80.62 | 45.97 |
| PGD-10+INT 8 | 66.77 | 39.57 |
| PGD-10+INT 4 | 66.20 | 37.75 |
| PGD-10+INT 2 | 55.34 | 29.36 |

