# OpenReview forum: "Non-uniform Noise Injection For Enhancing DNN Adversarial Robustness And Efficiency"
_ICLR.cc/2024/Conference — Submitted to ICLR 2024_

### Official Review · Reviewer_4ReP · 2023-10-25

**Soundness:** 2 fair
**Presentation:** 2 fair
**Contribution:** 2 fair
**Rating:** 3
**Confidence:** 5

**Summary:**

This paper introduces a novel method to address the challenges of adversarial attacks and computational costs in Deep Neural Networks (DNNs). Unlike previous studies that uniformly inject noise for robustness, this method strategically applies non-uniform noise at each DNN layer to disrupt adversarial attacks while preserving essential neurons. Experimental results show that this approach effectively enhances both robustness and efficiency in various attack scenarios, model architectures, and datasets.

**Strengths:**

The method of finding core and non-core neurons while simultaneously using those values to introduce noise is a good approach, in my opinion.

**Weaknesses:**

* It is stated that "A neuron is regarded as essential if its approximation from ˜z is larger than the predefined threshold." However, I am unsure why this makes it an important neuron. ˜z is used to reduce the value error through the MSE of z, meaning that the larger the z value, the more it will affect the size of ˜z. Therefore, an explanation is needed on why neurons with larger ˜z values are considered important.
The experiment tested both top-K injection and N:M injection methods, but there was no experiment that simply added noise values to a random N% of neurons. Therefore, there is a lack of validity in claiming that this method is effective.

* During the experimental process, in the experiment where noise was added to non-core neurons, I think there needs to be a control group that adds different noise to those neurons as a comparison to the approximated values used for detection. It is necessary to verify whether the method of injecting this noise actually contributes to performance improvement or simply enhances computational efficiency.

* Too much limited and insufficient experiments: There are no state-of-the-art attack such AutoAttack [1] and no state-of-the-art defense baselines such as AWP [2], SCORE [3], and ADML [4]. In addition, based on ADML, not only CNN structure and Transformer structures should be needed to validate.

---
References

[1] Croce, Francesco, and Matthias Hein. "Reliable evaluation of adversarial robustness with an ensemble of diverse parameter-free attacks." International conference on machine learning. PMLR, 2020.

[2] Wu, Dongxian, Shu-Tao Xia, and Yisen Wang. "Adversarial weight perturbation helps robust generalization." Advances in Neural Information Processing Systems 33 (2020): 2958-2969.

[3] Pang, Tianyu, et al. "Robustness and accuracy could be reconcilable by (proper) definition." International Conference on Machine Learning. PMLR, 2022.

[4] Lee, Byung-Kwan, Junho Kim, and Yong Man Ro. "Mitigating adversarial vulnerability through causal parameter estimation by adversarial double machine learning." Proceedings of the IEEE/CVF International Conference on Computer Vision. 2023.

**Questions:**

Refer to Weaknesses

---

> ### Author Response · Authors · 2023-11-22
>
> Thanks for your valuable suggestions and comments. Details can be found below.
>
> **(1) Why neurons with larger ˜z values are considered important.**
>
> Intuitively, activations with a larger magnitude have more impact on overall model inference; hence important. We use the approximate values as proxy to choose neurons with large magnitude.
>
> **(2) Add noise values to a random N\% of neurons. And control group that adds different noise to those neurons as a comparison to the approximated values used for detection.**
>
> This is part of the ablation experiment that we missed to support our claim more rigorously, and we will include this experiment in future enhancements.
>
> **(3) Insufficient experiments.**
>
> Your constructive comments are much appreciated. We will include a comparison of these robustness-related methods in future experiments.

---

### Official Review · Reviewer_xfb1 · 2023-10-31

**Soundness:** 3 good
**Presentation:** 3 good
**Contribution:** 3 good
**Rating:** 6
**Confidence:** 4

**Summary:**

The paper studies the use of a non-uniform noise injection method to improve adversarial robustness and efficiency. The proposed method first trains a linear layer-wise approximation of the layer's output and uses the approximation to determine which neurons are essential and which are not. A binary mask is used to replace the outputs of non-essential neurons with their approximations. The paper later proves that this process is unlikely to decrease clean accuracy. The improvements in adversarial robustness and computational efficiency are also discussed. The experiments further validate these claims.

**Strengths:**

1. The paper is well-organized and easy to understand.
2. The proposed method that uses non-uniform noise injection is novel and produces promising results.
3. The discussion provides theoretical support for the validity of the proposed methods.

**Weaknesses:**

1. The threshold and the proportion of non-essential neurons should also be reported. It is also helpful to provide the distribution of the difference between $z$ and $\tilde{z}$, and study the relationship between accuracies and the choice of threshold.

2. The theorem does not seem to directly link to the results. Based on the theorem, the clean accuracy can be higher than that without the noise injection, and it does not explain the consistent minor drop in clean accuracy.

**Questions:**

1. The learning of $\tilde{W}$ and $\tilde{b}$ minimizes the MSE. Is it possible that the optimal learned solution is $W=\tilde{W}P$ and $b=\tilde{b}$?
2. Are there any specific reasons that $\epsilon=4/255$ in Table 1 while $\epsilon=8/255$ in Table 3?

---

> ### Author Response · Authors · 2023-11-22
>
> We appreciate you for the positive feedback and your recognition of our contribution. Your questions are truly insightful which merit further investigation. Our answers are listed as follows:
>
> **(1) The threshold and the proportion of non-essential neurons.**
>
> It indeed was a writing mistake. The threshold is not predefined, it is found during the top-k process. This threshold is the smallest number in the top-k, and the purpose of the threshold is to facilitate the generation of the mask. We will revise the manuscript to avoid ambiguity.
>
> **(2) The distribution of the difference between z and ~z.**
>
> Thank you for your valuable advice. We will take this into consideration in our future improvements to explore the underlying reasons and to support our approach even more.
>
> **(3) Theorem explanation.**
>
> Our theoretical proof is divided into two parts, the first part is to show that this type of noise injection is naturally brought about by the dimensionality reduction method **random projection**, which has no effect on the distance between vectors and vectors in the high-dimensional space and therefore has no effect on the model's performance in clean accuracy. The second part shows that this kind of noise injection can be regarded as a kind of Gaussian noise, which improves the robustness, but too much noise injection affects the performance of the model in clean accuracy. Therefore, we chose the non-uniform noise injection that embodies a kind of trade-off.
>
> **(4) Optimal learned solution of approximation weight and bias.**
>
> Thank you for your careful consideration. Essentially, this approach is slightly different from the optimization approach using MSE, mainly is that the difference in the optimization objective has an impact on the convergence of the model training, and the way MSE is chosen is a single objective optimization, which makes the model converge better. If choosing weights and biases, the optimization objective becomes four, and the hyperparameters are more difficult to choose. In addition, we want to make the activation having little influence the result, so we do the optimization in this way. We will try the method you mentioned.
>
> **(5) Adversarial attack epsilon choice.**
>
> In Table 1, this validation of adversarial robustness does not use adversarial training, which means that the neural network is not pre-fitted on the adversarial samples, and thus in case of larger attack intensity epsilon, the robustness accuracy of both our method and the original model will be very low, which does not reflect the enhancement of our model in adversarial robustness. Instead, Table 3 is trained for confrontation and can withstand higher intensity attacks epsilon(e.g. 8/255).

---

### Official Review · Reviewer_A4cN · 2023-10-31

**Soundness:** 2 fair
**Presentation:** 1 poor
**Contribution:** 2 fair
**Rating:** 3
**Confidence:** 2

**Summary:**

This paper proposed a novel method that inject non-uniform noise to non-essential neurons so that the adversarial robustness of the network is enhanced while the clean accuracy is not harmed.

**Strengths:**

The work deals with a significant and inspiring topic: enhancing adversarial robustness and efficiency simultaneously.

**Weaknesses:**

- I find the paper a little hard to follow. Specifically, I cannot determine what “irregular 50%” and “structured/unstructured” mean in the paper’s context. It is very likely because I don’t have sufficient background knowledge and I would appreciate it if the authors could refer me to 2-3 most related (and preferably recent) papers that can help me gain the basics. But for now, I tend to believe the authors fail to make the paper easy to follow.

- AutoAttack is not included in threat models, which I believe is necessary to show the adversarial robustness of a new method.

- The performance boost of the proposed method seems to be limited. In Table 3, clean accuracies decreased by a noticeable margin with limited improvement in robustness accuracies. So, it is hard to say that a better accuracy-robustness trade-off is achieved. Also, more baseline AT methods (e.g. TRADES) would be helpful to show the method’s effectiveness.

- As it is claimed that the proposed method enhances adversarial robustness and execution efficiency simultaneously, it is important to show directly how much the efficiency is improved directly (e.g., throughput) compared to the baselines.

**Questions:**

Can you intuitively explain why “a neuron is regarded as essential if its approximation from $\widetilde{z}$ is larger than the predefined threshold”?

---

> ### Author Response · Authors · 2023-11-22
>
> Thanks so much for your feedback on our methods and constructive comments on the validation of robustness. We provide the answers below.
>
> **(1) “irregular 50\%” and “structured/unstructured” mean in the paper’s context.**
>
> Thank you for your advice. We realize that the lack of explanation of sparsity can be troubling to readers. So we need to explain the irregular as well as the structured/unstructured in the manuscript. `irregular/unstructured' and 'structured' both can be seen as a kind of sparsity. ‘irregular/unstructured’ means that the **activation pattern** after sparsification (top-k) is irregular[1]. 'structured' means for a fixed block, the sparsity is determediated[1] [2]. It's like the one shown in Figure 2 in the text. The difference between the two forms of sparsity is mainly in the efficiency of hardware execution; structured sparsity is more efficiently executed due to the fact that it can be chunked for the extraction of deterministic sparse values.
>
> **(2) Autoattack to show the adversarial robustness.**
>
> The autoattack method is the state-of-the-art method in recent years, in our experiments we ignored this method because of insufficient research, in the future experiments we will add the validation of this method, thank you for your suggestion.
>
> **(3) The performance limitation.**
>
> We noticed this limitation, the problem should be in the search of the optimal hyperparameters for adversarial training, for our model, the optimal hyperparameters may not be found, which leads to overfitting of our model during adversarial training. In our future work, we will use standard adversarial training[3] to prevent overfitting in order to further evaluate the robustness.
>
> **(4) Efficiency criteria.**
>
> We will consider more measures that indicate model efficiency.
>
> **(5) A neuron is regarded as essential if its approximation is larger than the predefined threshold.**
>
> This was a writing mistake that causes confusion. The threshold is not predefined, it is found during the top-k process. This threshold is the smallest number in the top-k, and the purpose of the threshold is to facilitate the generation of the mask. We will revise the manuscript to avoid ambiguity.
>
> References:
>
> [1] Ji Y, Liang L, Deng L, et al. TETRIS: Tile-matching the tremendous irregular sparsity[J]. Advances in neural information processing systems, 2018, 31.
>
> [2] Nvidia, “Nvidia a100 tensor core gpu architecture.” 2020. [Online]. Available: https://images.nvidia.com/aem-dam/en-zz/Solutions/data-center/nvidia-ampere-architecture-whitepaper.pdf
>
> [3] Leslie Rice, Eric Wong, J. Zico Kolter: Overfitting in adversarially robust deep learning. ICML 2020.

---

> > ### Comment · Reviewer_A4cN · 2023-11-23
> >
> > I thank the authors for replying to my concerns. However, I think too much work is left for the future and the current version is not ready to publish in this venue. After I read other reviews and the authors' responses, I decided to lower my rating to 3.

---

### Official Review · Reviewer_UxUE · 2023-11-04

**Soundness:** 3 good
**Presentation:** 2 fair
**Contribution:** 2 fair
**Rating:** 3
**Confidence:** 4

**Summary:**

This work suggests that only a subset of neurons is critical in representation learning and proposes a selection method to categorize neurons into essential neurons and non-essential neurons. Subsequently, the authors apply non-uniform noise injection to these two types of neurons in order to improve adversarial robustness and maintain clean accuracy.

**Strengths:**

1. The hypothesis that only a subset of neurons are critical in representation learning, and the rest can tolerate noise perturbations without affecting performance, is interesting.
2. Propose a method for distinguishing between essential neurons and non-essential neurons.
3. Propose a non-uniform noise injection method tailored for essential neurons and non-essential neurons.

**Weaknesses:**

1. Limited novelty: The proposed method is a marginal improvement on existing methods.
2. Over-claimed contribution: In the experiments, the proposed method still leads to a decrease in clean accuracy, rather than truly "retaining clean accuracy" as claimed.
3. Robustness evaluation is inaccurate: It is suggested that the authors employ the experimental setup for the standard adversarial training [1] and use AutoAttack [2] for assessing the model's robustness. Compare the effectiveness of the proposed method on the model's best robustness and last robustness.

[1] Leslie Rice, Eric Wong, J. Zico Kolter: Overfitting in adversarially robust deep learning. ICML 2020.
https://github.com/locuslab/robust_overfitting
[2] Francesco Croce, Matthias Hein: Reliable evaluation of adversarial robustness with an ensemble of diverse parameter-free attacks. ICML 2020.
https://github.com/fra31/auto-attack

**Questions:**

If the proposed method can substantially achieve higher robustness compared to standard adversarial training, I will increase my score.

---

> ### Author Response · Authors · 2023-11-22
>
> We greatly appreciate your constructive comments. We will include the mentioned robustness validation methods in future experiments. We will incorporate your comment on the novelty and contribution with more rigours analysis.

---

### Meta-Review · Area_Chair_DB82 · 2023-12-08

**Metareview:**

This paper introduces non-uniform noise into DNN layers to disrupt adversarial perturbations, thereby enhancing the adversarial robustness of DNNs.

During the rebuttal, the authors did not well address the concerns raised by three reviewers, including technical novelty, limited performance, lack of baseline, and insufficient evaluation (especially without using AutoAttack). Thereby, the AC recommends rejecting this paper.

**Justification For Why Not Higher Score:**

The authors have not addressed the major concerns raised by three reviewers, including technical novelty, limited performance, lack of baseline comparisons, and insufficient evaluation.

**Justification For Why Not Lower Score:**

NA

---

### Decision · Program_Chairs · 2024-01-16

Reject